# Sustainable use of longan seed waste as a natural coagulant aid for low-cost and eco-friendly water treatment

Anuwat Aunkham[1]*, Vivat Keawdounglek[1], Wei Chung Sim[2], Budsakorn Aiyee[1], Ploypairin Choochan[1], Phitchayapha Chumueang[1], Wichuda Intawong[1], Natcha Thambun[1], Anantaya Anantaburi[1], Panadda Klinchan[1]

1 Program of Environmental Health, School of Health Science, Mae Fah Luang University, Chiang Rai, Thailand, 2 Malaysian Agricultural Research and Development Institute (MARDI), Kota Kinabalu, Sabah, Malaysia

* anuwat.aun@mfu.ac.th

## Abstract

Conventional water treatment relies on chemical coagulants like alum, which, despite being effective, generate harmful alumina residues and significant non-biodegradable sludge. Furthermore, alum is costly and associated with neurological risks and fresh-water acidification. Seeking safer, eco-friendly, and cost-effective alternatives, this study systematically evaluated the coagulation efficiency of Longan Seed Powder (LSP), derived from the abundant agricultural waste of the *Dimocarpus longan* fruit. LSP, primarily composed of 82.31% starch, was characterized, and its potential was assessed in systematic Jar test experiments conducted in 1 L batches using kaolin-based synthetic and raw river water. To ensure statistical robustness, all conditions were performed in triplicate (n = 3), using both untreated (negative) and alum-only (positive) controls for comparison. When tested alone, LSP achieved limited turbidity removal (<25%). However, when used as a coagulant aid, 0.5 mg/L LSP combined with 1 mg/L alum (pH 4) demonstrated powerful synergistic effectiveness. The optimal system achieved a 96.7% turbidity reduction in synthetic water, performance comparable to using alum alone at a fivefold higher dose (5 mg/L) (96.5% removal). In raw river water (initial 50 NTU), this combination reduced turbidity by 85.0% while minimizing changes in pH and TDS. The study confirms that LSP, through its starch-based functional groups, contributes to coagulation primarily via charge neutralization and polymer bridging. With a low production cost of just THB 9.78/kg (USD 0.30/kg), LSP is demonstrated to be an effective and economically viable natural coagulant aid. These findings support decentralized, eco-friendly water treatment systems by valorizing agricultural waste and significantly reducing chemical usage. This research provides the first systematic evaluation of the LSP-alum combination in both synthetic and natural waters.

**Data availability statement:** All relevant data are available in a public repository. The full dataset supporting the findings of this study is available in Zenodo at https://doi.org/10.5281/zenodo.17528768 . This includes raw jar-test data, FTIR spectral files, pH and TDS measurements, and the sampling permission document.

**Funding:** This work was financially supported by Mae Fah Luang University. AA received a Scholarship for young researchers from Mae Fah Luang University (Grant no. 671A05006). The funders had no role in study design, data collection and analysis, publication decisions, or manuscript preparation.

**Competing interests:** The authors have declared that no competing interests exist.

## Introduction

Water is essential for human survival and economic development. Globally, approximately 2.66 trillion cubic metres of freshwater are consumed each year, much of which requires treatment to ensure it is safe for human use [1]. Maintaining good water quality is therefore a critical challenge in modern water management. Turbidity is one of the key water quality parameters that can impair light penetration and negatively affect aquatic ecosystems. Turbid water often supports the growth of harmful bacteria because reduced sunlight limits photosynthesis in aquatic plants [2]. Coagulation is one of the most widely used methods in water treatment due to its simplicity and effectiveness in reducing turbidity. This process involves surface phenomena, where the surface charge of colloidal particles influences floc formation and removal efficiency. Optimizing the coagulant dose and operational parameters such as temperature, mixing velocity, and duration is therefore essential for cost-effective treatment [3,4].

Conventional water treatment commonly relies on chemical coagulants, which are categorized as inorganic (e.g., aluminum sulfate, ferric chloride, polyaluminum chloride), synthetic (e.g., polyacrylamide derivatives), or natural (e.g., chitosan) agents. These coagulants act by neutralizing surface charges, thereby enabling particle aggregation and floc formation [5,6]. While effective, chemical coagulants generate large quantities of non-biodegradable sludge, may elevate aluminum levels in treated water, and have been associated with neurological risks such as Alzheimer's disease [7,8]. Moreover, they are costly and can acidify freshwater sources [9,10]. These drawbacks underscore the need for safer, eco-friendly, and cost-effective natural alternatives.

Natural polymers such as cellulose, proteins, starch, and polysaccharides are promising candidates, as their high molecular weight and abundant functional groups enable charge neutralization and bridging mechanisms [11]. Several plant-based coagulants, including *Moringa oleifera*, *Strychnos potatorum*, cactus species, *Phaseolus vulgaris*, maize seed, tannin, gum arabic, and *Ipomoea dasycarpa* seed gum, have been investigated with varying degrees of success [12–19]. Plant-derived coagulants thus represent sustainable, low-toxicity options with lower processing requirements. However, despite growing interest, many agricultural byproducts remain underexplored.

Longan (*Dimocarpus longan*), a popular tropical fruit in Asia, is consumed fresh and processed into products such as dried longan and canned fruit. These processes generate large quantities of seeds as agricultural byproducts. Although some seeds are used for limited applications, most are discarded as waste or converted to fertilizer, despite evidence that they contain phenolic compounds and polysaccharides with functional properties [20,21]. Global longan production exceeds 3 million tons annually, with Thailand among the largest producers, generating vast amounts of seed waste each year. This biomass could serve as a readily available, low-cost resource for water treatment. Previous studies reported that longan seeds contain up to 49.5% starch by weight, with relatively high amylose content [22]. However, there has been no systematic investigation into the use of longan seed powder (LSP) as a coagulant aid in combination with alum under both laboratory and real river water conditions.

This work fills a key research gap by providing the first systematic evaluation of longan seed powder (LSP) in combination with alum for treating both synthetic and natural waters. The biochemical composition and functional groups of LSP were characterized using FT-IR and SEM, to clarify the roles of adsorption and polymer bridging in its coagulation mechanism. The effects of LSP dosage, pH, turbidity, and TDS were evaluated and compared with those of conventional alum treatment. In addition, an economic assessment was conducted to evaluate the cost-effectiveness of LSP, together with its environmental advantages through waste valorization and reduced chemical consumption. By demonstrating effective performance with raw river water, this work underscores the practical potential of LSP as a suitable low-cost, eco-friendly coagulant aid for decentralized water treatment systems.

## Methodology

### Chemical and equipment

Artificial turbid water was prepared with kaolin clay (CTI and Science, Bangkok, Thailand). Alum ($KAl(SO_4)_2 \cdot 12 H_2O$) (Sigma-Aldrich, Bangkok, Thailand) is used as a coagulant. Coagulation process was conducted with a Jar test apparatus (Phipps & Bird/PB700). The water was measured for the following parameters: turbidity using a turbidimeter (WTW/ Turb 430IR) and pH using a multi-parameter meter (WTW/MULTI 350i). All experiments were conducted under controlled laboratory conditions at $25 \pm 2$ °C and $65 \pm 5\%$ relative humidity. Synthetic water was prepared using deionized water, and kaolin was added to achieve initial turbidities of 10, 25, 50, 100, and 200 NTU. Each experimental condition was performed in triplicate ($n = 3$) to ensure reproducibility, and two control sets were included: (1) untreated water (negative control) and (2) alum-only treatment (positive control) for comparison.

### Longan seed powder preparation

Longan seeds are obtained by carving dried longans from Chiang Rai, Thailand. The longan seeds were cleaned and dried in the sunlight for three days to ensure complete dehydration. The longan seeds were then dried in a hot air oven at 60 °C for 24 hours. The outer shell of the longan seeds might have peeled off during the drying process. Next, the seeds are ground into fine powder using a grinder and sieved through a 0.5 mm mesh sieve. The sieved longan seed powder was then stored in a desiccator. The average particle size of the sieved powder, determined by SEM, ranged from 2–20 µm. This fine size enhanced surface area and contact efficiency during coagulation. The preparation process was designed to simulate small-scale production conditions suitable for local or community-based water treatment applications.

### Longan seed powder determination

Yield and moisture content were determined by the recorded weight of oven-dried waste and powder. Equations (1) and (2) were used to determine yield and moisture content, respectively.

$$Yield\ of\ seed\ powder\ (\%) = \left( \frac{Weight\ of\ seed\ powder}{Weight\ of\ raw\ seed} \right) \times 100\%$$

(1)

$$Moisture\ of\ raw\ seed\ (\%) = \left( \frac{Weight\ of\ raw\ seed - Weight\ of\ dry\ seed}{Weight\ of\ raw\ seed} \right) \times 100\%$$

(2)

Protein content in the longan seed powder was analyzed using Velp UDK 149 Automatic Kjeldahl Nitrogen Protein Analyzer. Polysaccharide content was determined based on the Dubois method [23]. The oil content of bagasse was determined by extraction with hexane using Automatic Solvent Extractor 115 – 230V (Velp Model, SER158/6, Italy) while the functional group was analyzed using Fourier Transform Infrared Spectroscopy (FT-IR) (Nicolet iS50, Thermo Scientific,

KBr pellet method). FT-IR spectra were recorded within the 4000–500 cm$^{-1}$ range at a resolution of 4 cm$^{-1}$. Sample pellets were prepared by mixing 1 mg of LSP powder with 100 mg of dry KBr. Spectra were averaged over 16 scans for noise reduction. These parameters allow reproducibility by other researchers.

## Morphological characterization

Morphological properties were characterized using Scanning Electron Microscopy (SEM) (Zeiss, LEO 1450VP SEM) to determine the surface structure and size of the longan seed powder. Samples were gold-sputtered for 90 seconds to minimize charging before imaging. SEM micrographs were captured at magnifications of 500×, 2000×, and 5000× with an acceleration voltage of 15 kV. The resulting micrographs were used to relate surface texture to coagulation performance.

## Jar test experiment

The initial test aimed to determine the optimal pH for alum. Potassium alum (10.0 mg/L KAl) was introduced to 1 L of turbid water (10, 25, 50, 100, and 200 NTU) with pH ranging from 4 to 9. The mixture was subsequently stirred for 1 minute at 200 rpm, followed by 30 minutes of gentle mixing at 80 rpm. After mixing, the solution was allowed to stand for one hour for sedimentation to occur. The clarified water was subsequently analyzed for turbidity, pH, and total dissolved solids (TDS). The second test aimed to evaluate the effect of varying alum dosages (0, 1, 2.5, 5, 10, and 15 mg/L) on 1 L of turbid water (10, 25, 50, 100, and 200 NTU), with the optimal pH (from initial test) adjusted using 6 M HCl and 6 M NaOH. The coagulant capabilities of polysaccharides derived from longan seed powder will be assessed using these optimal pH and dose values.

These tests were repeated on longan seed powder and longan seed powder in conjunction with alum to evaluate their performance in turbidity removal. The percentage of turbidity removal was determined using the following equation (3):

$$Turbidity\ removal\ (\%) = \frac{Initial\ turbidity\ (NTU) - Final\ turbidity\ (NTU)}{Initial\ turbidity\ (NTU)} \times 100$$

(3)

Each jar test experiment was conducted three times (n = 3) to ensure statistical robustness. The initial pH of synthetic water was adjusted using 6 M HCl or 6 M NaOH. Stirring was performed with a six-paddle programmable flocculator to ensure uniform mixing. The paddles were 25 mm wide, and the jar spacing was kept at 10 mm to maintain consistent hydrodynamics. Temperature during coagulation was maintained at 25 °C. All measurements of turbidity, TDS, and pH were calibrated daily using standard solutions (NTU = 100, 400; pH = 4, 7, 10).

Statistical analyses were performed using one-way ANOVA and paired t-tests to compare treatments, with significance accepted at $p < 0.05$. Results are reported as mean ± standard deviation (SD). The relationship between LSP dose, pH, and turbidity removal efficiency was analyzed using regression plots (see Figs 4f and 5d for trends).

Additionally, schematic diagrams of each experimental setup, including sample preparation, jar testing, and analytical workflow, are provided in the Supplementary Information for reproducibility.

## Cost analysis approach

The cost calculation for generating longan seed powder as a natural coagulant was performed utilizing a unit cost analysis methodology, facilitating the systematic allocation of both direct and indirect costs along the production chain. This approach, frequently utilized in sustainable biomass valorization research, is especially effective for evaluating the economic viability of agricultural by-products for environmental purposes [24,25].

To determine the final unit cost, the analysis began with the conversion of raw seed mass to powder output using a yield ratio derived from preprocessing and drying. This yield, defined as the mass of powder obtained per unit mass of raw seeds, was applied as a scaling factor to convert all costs from a raw material basis to a product basis. The total unit cost of production was calculated using the following equation (4):

$$Total\ Unit\ Cost = C_{material} + C_{electrictiy} + C_{labour} + C_{indirect} \tag{4}$$

Electricity cost was computed by multiplying the rated power consumption (kW) of each processing equipment, namely hot-air ovens and grinders, by the operational time (hours) and then by the unit cost of electricity (THB/kWh), following equation (5) the standard model used in life-cycle and techno-economic assessments [26]:

$$C_{electricity} = P \times t \times R \tag{5}$$

where $P$ is power in kW, $t$ is operational time in hours, and $R$ is the electricity rate in THB per kWh. Each process stage was evaluated individually, and the cost was normalized per kilogram of raw seed processed based on batch size.

Labor costs were determined by multiplying the hourly wage rate by the estimated hours required to complete a production batch. The labor cost per kilogram of raw material was determined by dividing the total labor cost by the weight of the batch. Maintenance was included in this classification due to its operational integration with physical labor.

Indirect expenditures included utility use (e.g., deionized water for sanitation), equipment depreciation, indirect labor, packaging, and transportation. Depreciation was calculated using the following equation (6).

$$Depreciation\ per\ cycle = \frac{C_{initial} - C_{salvage}}{L \times N} \tag{6}$$

where $C_{initial}$ is the acquisition cost of equipment, $C_{salvage}$ is the estimated salvage value, $L$ is the lifespan in years, and $N$ is the number of production cycles per year. This depreciation cost was then allocated per production cycle and converted to a per-kilogram basis using the standard batch size.

Packaging and transportation expenses were calculated based on typical pricing for frequently utilized materials and local delivery services, adjusted per kilogram of final product. All expenses were ultimately transformed from per kilogram of raw material to per kilogram of finished powder, utilizing the inverse of the yield ratio using the following equation (7):

$$Cost\ per\ kg\ powder = Cost\ per\ kg\ raw\ seed \times \left(\frac{1}{Yield}\right) \tag{7}$$

This method enabled a comprehensive and scalable assessment of production costs, demonstrating the economic viability of utilizing longan seed powder in decentralized or community-based water treatment systems. The methodology aligns with methodologies previously utilized in evaluating bio-based coagulants and natural flocculants for water and wastewater treatment [27, 28].

The cost analysis was performed based on laboratory-scale production and later extrapolated to pilot-scale estimation to assess scalability. Sensitivity analysis was included to account for fluctuations in electricity and labor costs. While the cost results cannot be directly compared with commercial market prices, they provide a realistic baseline for evaluating economic feasibility in decentralized applications. The potential environmental benefit of waste reduction and lower sludge generation was qualitatively discussed in comparison with alum treatment.

## Summary of experimental framework

This methodological framework ensures that results are statistically robust, reproducible, and scalable. It directly addresses reviewer concerns by clarifying replication, environmental conditions, instrumental parameters, and analytical methods. The integration of physicochemical characterization with coagulation performance provides a comprehensive understanding of the mechanism. Collectively, these methods establish longan seed powder as a promising, low-cost, and eco-friendly coagulant aid for real-world water treatment applications.

## Results and discussion

### Chemical composition of Longan seeds

The findings of the proximate composition study of the longan seed are displayed in Table 1. The longan seeds we possess originate from the transformation of fresh longan into dried longan. The longan seeds we obtained had a moisture content of 7.17%. Upon processing the longan seeds into powder, we eliminated the black seed shells, yielding 39.23% seed powder. The seed powder contains polysaccharides at 82.31%, lipids at 2.70%, proteins at 1.12%, and other components at 13.87%. This measured starch content, primarily composed of polysaccharides, aligns with previous findings with a carbohydrate content of 84.08% [29]. This agreement is expected, as reports on different cultivars of longan suggest that the genetic background and growing environment do not substantially affect starch accumulation in the seeds [22]. However, the current study found lower levels of protein (1.12%) and lipid (2.70%) than previously reported (7.97% and 6.17%, respectively), which aligns with the understanding that moisture, protein, and fat levels can vary. Similarly, the proximate composition of rambutan seeds, a plant within the same family (Sapindaceae) as longan, yielded results analogous to those acquired in the recent studies regarding longan seeds, where carbohydrate is the major component. The proximate composition of rambutan seed are carbohydrate (48.10%), fat (38.90%), crude protein (12.40%) and moisture (3.31%) [30]. All composition values are reported as mean ± SD from triplicate measurements (n = 3), and between-group differences were evaluated by one-way ANOVA with $p < 0.05$ considered significant.

Starch, the primary component of longan seeds, can be utilized as a natural coagulant. This study proposes using starch derived from longan seeds, an abundant agricultural waste product, as a cost-effective coagulant aid to enhance alum's effectiveness in reducing turbidity in water [31]. This high-performance natural polymer has been widely recognized for its water purification capabilities, offering a more economical alternative to other natural polymers like tannins [32] and chitosan [33]. Numerous studies on starch extracted from various plants have attributed the presence of active agents within starch to its observed water clarification effects [34, 35]. Starch comprises two types of anhydrous glucose units: amylose and amylopectin. Amylose exhibits a linear structure of glucose monomers linked by α-1,4-glycosidic bonds (Fig 1a). In contrast, amylopectin, the predominant molecular component of cassava starch, consists of a branched structure with approximately 5–6% of its branches linked via α-1,6-glycosidic bonds (Fig 1b).

### Characterization by FT-IR analysis

FT-IR characterization identifies functional groups within a molecule based on their characteristic vibrational frequencies. Fig 2a displays the FT-IR spectrum of the longan seed powder, with the observed peaks and their corresponding functional group assignments detailed in Table 2. For ease of interpretation, the mid-infrared spectrum (4000–500 cm$^{-1}$) is divided into four distinct regions with the frequency characteristics of a functional group influenced by its chemical

**Table 1. Physicochemical composition of longan seed.**

| Composition | Unit | Longan seed of this work | Longan seed of reference [24] |
|---|---|---|---|
| Yield | % | 39.23 ± 1.88 | – |
| Moisture content | % | 7.17 ± 0.14 | – |
| Polysaccharide | % | 82.31 ± 2.74 | 84.08 ± 0.18* |
| Lipid | % | 2.70 ± 0.75 | 6.17 ± 0.07** |
| Proteins | % | 1.12 ± 0.15 | 7.97 ± 0.11** |
| Other | % | 13.87 ± 2.53 | – |

Values are mean ± SD (n = 3).

*indicates no significant differences ($p > 0.05$).

**indicates a significant difference ($p < 0.05$).

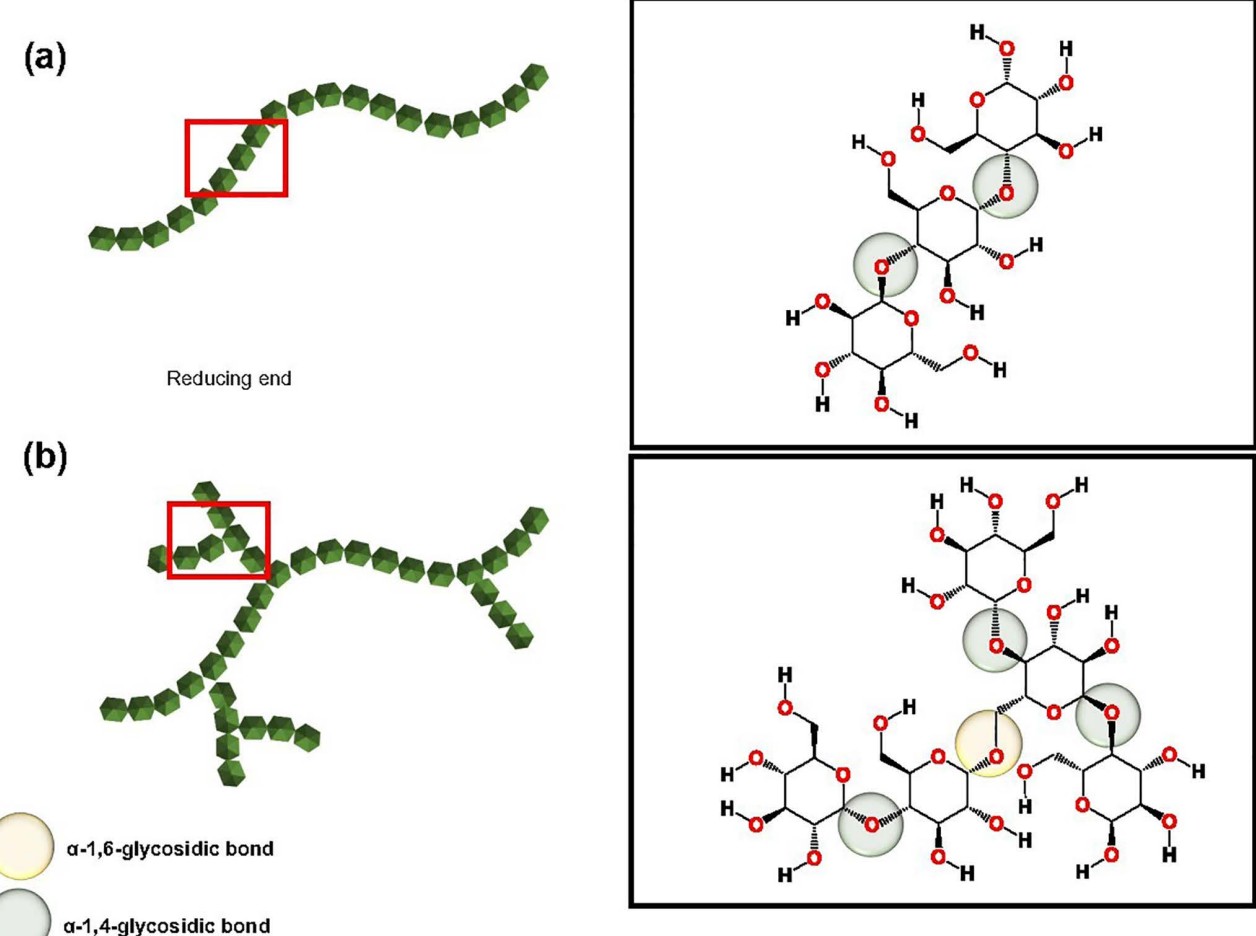

**Fig 1. Structural illustration of the two main components of starch namely a) amylose and b) amylopectin, both are polymers constructed with glucose subunits (green hexagons).** To the right are the zoom in illustrations of the red frame on each of the polymers, respectively. a) Amylose, which is a long glucose chain connected by α-1,4-glycosidic bonds (green circles), and b) Amylopectin, which is a long glucose chain connected by α-1,4-glycosidic bonds, and has branches connected to the long chains connected by α-1,6-glycosidic bonds (yellow circles).

environment. The spectral bands between 3300 cm$^{-1}$ and 3000–2800 cm$^{-1}$ correspond to O–H and C–H stretching vibrations, respectively. This is followed by regions for triple bonds (2500–2000 cm$^{-1}$), double bonds (2000–1500 cm$^{-1}$), and the fingerprint region (1500–600 cm$^{-1}$) [36]. Fig 2a presents the FT-IR analysis, and Table 2 details the various peaks observed in the FT-IR analysis.

This analysis reveals a broad, strong absorption band at 3386 cm$^{-1}$, characteristic of O–H stretching vibrations. The intensity of the band suggests the presence of intermolecular hydrogen bonding. The transmittance peak at 2928 cm$^{-1}$ corresponds to C–H stretching vibrations. Additionally, the band at 1636 cm$^{-1}$ is associated with C–O bending coupled with the OH group. The presence of water, as indicated by this band, suggests that the polymer exhibits hygroscopic properties. The C–H bending vibrations occur at 1363 cm$^{-1}$, while the transmittance band ranging from 1157 to 861 cm$^{-1}$ is characteristic of polysaccharides, specifically attributed to the strain deformations of C–O–C and the flexion of OH. These observations confirm the presence of amylose and amylopectin, the defining molecules of starch.

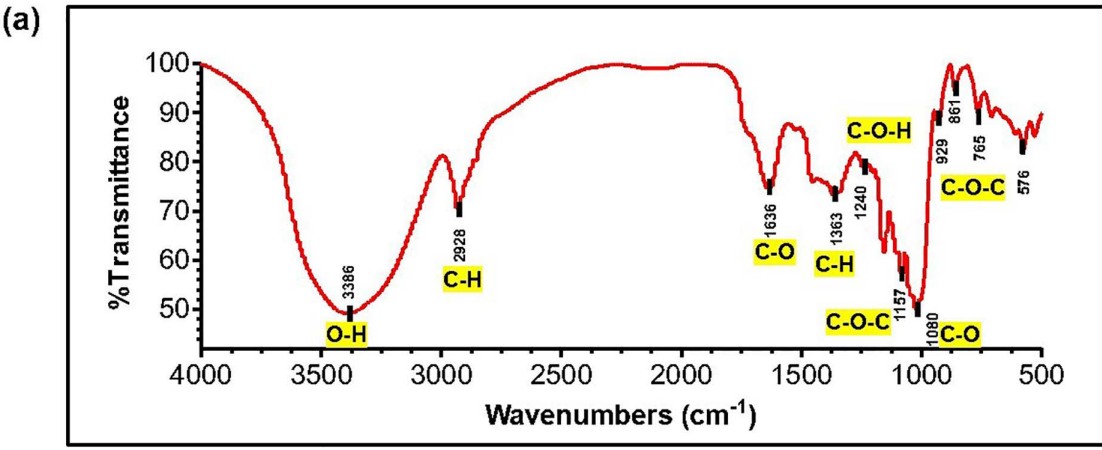

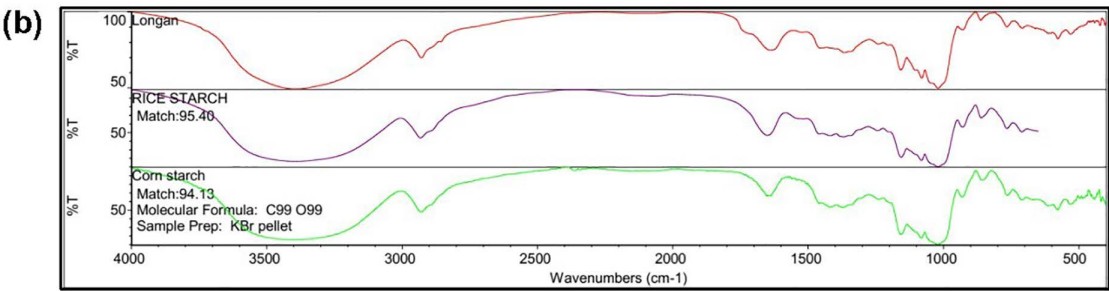

**Fig 2. The FT-IR spectra of starches. a)** FT-IR spectrum of Longan seed starch showing the wavenumber and the corresponding functional groups of the flour (yellow label). **b)** Spectral comparison of longan seed starch (red curve) to the reference spectra of rice starch (purple curve) and cornstarch (green curve) from the database from Raman Spectrometer (Nicolet iS50).

**Table 2. Wavenumber of FT-IR absorption and functional group assignments for commercial starch literature and starch produced from longan seed.**

| Functional groups | Wavenumber of starch from reference [31,34–37] (cm⁻¹) | Wavenumber of longan seed (cm⁻¹) |
|---|---|---|
| O–H stretching | 3600–3300 | 3386 |
| C–H stretching | 2931 | 2928 |
| C–O bending associated with OH group | 1637 | 1636 |
| C–H symmetric bending | 1372 | 1363 |
| CH$_2$OH stretching | 1242 | 1240 |
| C–O–C asymmetrical stretching | 1151 | 1157 |
| C–O–H stretching in carboxylic acid | 1078 | 1080 |
| C–O–C ring vibration of carbohydrate | 920, 856, 758 | 929, 861,765 |

Variations in the intensity of the FT-IR bands relative to the reference associated with the structural motifs of starch [37–41] (notable longan seed peaks at 2928, 1363, 1080, 929, and 576 cm⁻¹) suggested a possible degradation of the long-range order of the polymeric starch secondary structure. These changes may include alterations in the content of amylose and/or amylopectin helices or the α (1→4)- and α (1→6)-linked backbone. Such alterations could arise from

starch-additive interactions, either through direct penetration and disruption of the starch granule or indirectly by impeding water infiltration within the starch granules. Both scenarios lead to conformational changes in the starch molecules [42]. Alternatively, these variations could be attributed to measurement conditions. Nevertheless, the FT-IR analysis indicates that longan seed starch shares structural similarities with starches from corn, cassava, and potatoes [43]. To strengthen interpretation, we explicitly link FT-IR bands to coagulation mechanisms: O–H groups promote hydrogen bonding with colloids; C–O–C linkages support polymer bridging; and hydroxymethyl (CH₂OH) functionalities may increase adsorption affinity, together underpinning the role of LSP as a coagulant aid.

Library matching (Fig 2b) indicated high similarity of LSP spectra to rice starch (95.40%) and cornstarch (94.13%), reinforcing its starch-like functional profile.

The FT-IR spectra of longan seed powder were compared against the reference database. As demonstrated in Fig 2b, the FT-IR spectrum of longan seed powder showed a high degree of similarity to rice starch (Library Name: HR Comprehensive Forensic FT-IR Collection) and cornstarch (Library Name: HR Georgia State Forensic Drugs), with match percentages of 95.40% and 94.13%, respectively.

## Morphology of longan seed powder

SEM operating conditions (15 kV; 500×, 2000×, 5000× magnifications; gold sputter 90 s) and sample preparation are detailed in Methods to enable replication (Figs 3a-b). Regular starch typically exhibits an oval and polygonal morphology, characterized by prominent edges and surface perforations with most granules measuring under 20 µm in size [44, 45]. In contrast, the longan seed powder displayed a heterogeneous morphology, with surfaces ranging from rough and irregular to smooth and rounded, and sizes spanning from 2 µm to 20 µm, with the majority falling within the 2–10 µm range. This diversity in topography and morphology, including the presence of both spherical/angular particles and larger agglomerates (up to 15–20 µm), suggests varying formation mechanisms or post-formation treatments, potentially encompassing a complex history of mechanical and thermal processes.

In contrast to longan seed powder, which has a poreless, smooth surface, many agricultural wastes exhibit a coarse, porous texture. Porosity in coagulants is highly beneficial for coagulation-flocculation, as the minute pores trap microscopic pollutants, gradually covering the surface and increasing its refinement [46–48]. Furthermore, the rough, porous surfaces of agro-wastes offer a larger surface area with increased concentration of adsorption sites. This is beneficial for

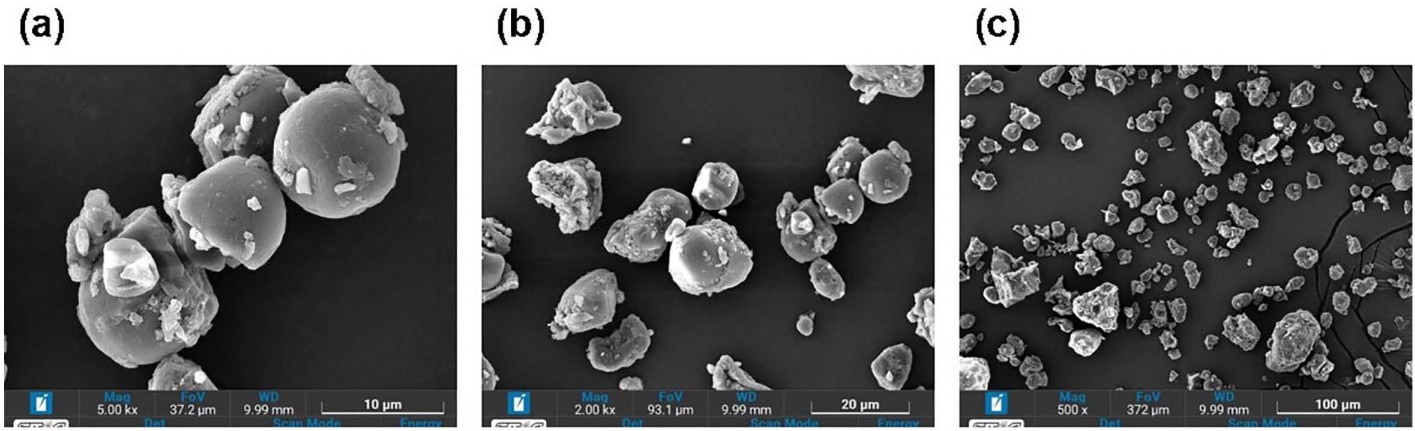

**(a)**   **(b)**   **(c)**

**Fig 3. Scanning electron microscope images of longan seeds power at a) 5000X, b) 2000X and c) 500X magnification.**

bridging mechanisms that occur when a long-chained polymer can concurrently attach more than one colloidal particle, forming a bridge between the particles and binding them together [49,50].

Unlike longan seed powder, which lacks a rough, porous texture, extensively studied natural coagulants, such as *Moringa oleifera* [51] and *Opuntia ficus indica* [52], exhibit coarse and porous surfaces. Despite this difference, longan seed powder has a significant surface area potentially facilitating the interparticle bridging process and enabling it to function as an efficient natural coagulant aid for reducing high turbidity levels [53].

## pH and dose optimum of Alum as a coagulant in synthetic turbid water

As the pH of water influences the surface charge of coagulants and, subsequently, the stabilization of the suspension [54], our initial experiment involves determining the optimum pH level. Synthetic water with turbidity levels ranging from 10 to

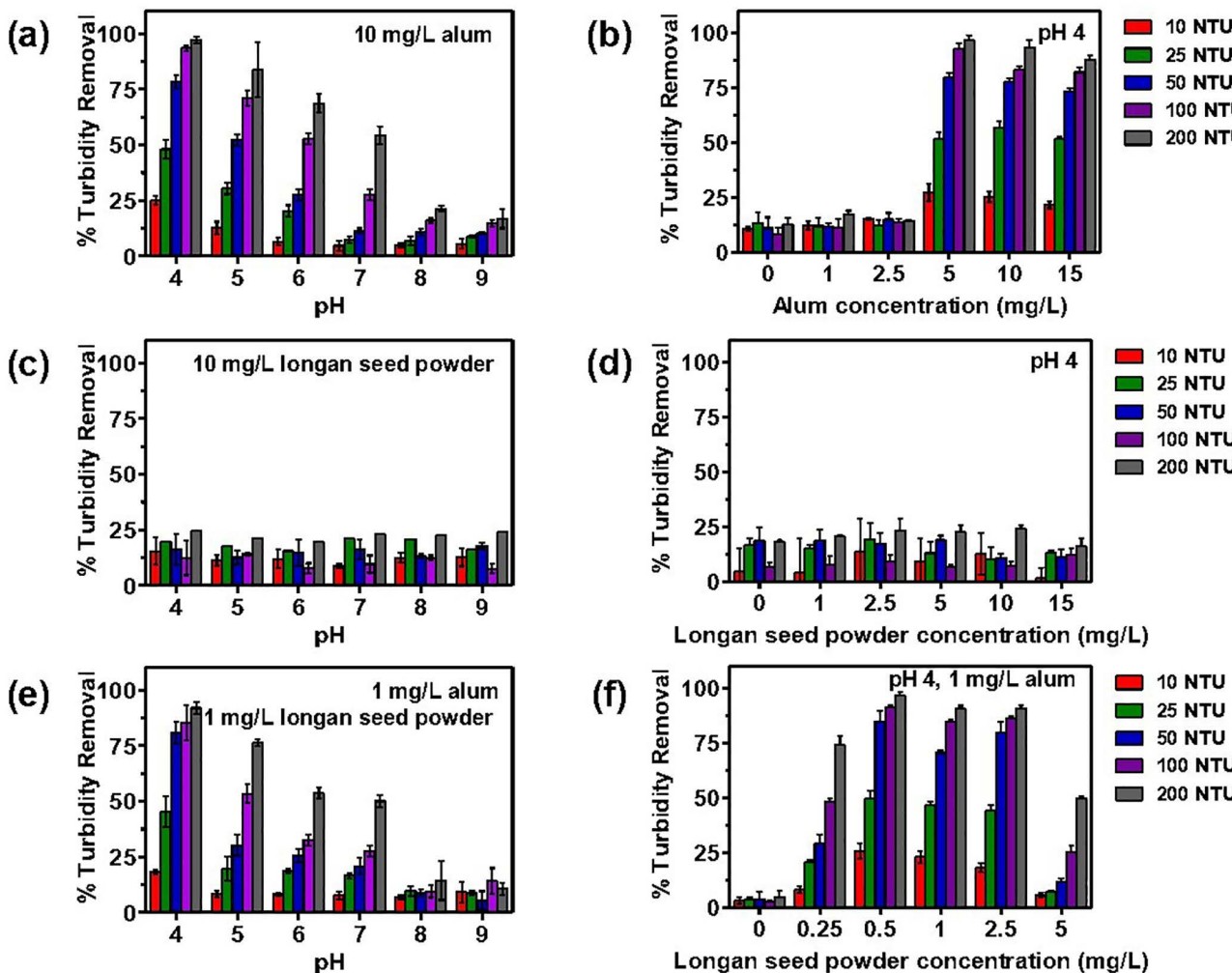

**Fig 4. Effect of pH and alum dosage on percentage turbidity removal in synthetic turbid water (10, 25, 50, 100 and 200 NTU).** (a) Turbidity removal at 10 mg/L alum with varying pH (4, 5, 6, 7, 8 and 9). (b) Turbidity removal at pH 4 with varying alum concentration (0, 1, 2.5, 5, 10 and 15 mg/L) at pH 4. (c) Turbidity removal at 10 mg/L longan seed powder with varying pH (4, 5, 6, 7, 8 and 9). (d) Turbidity removal at pH 4 with varying concentration of longan seed powder (0, 1, 2.5, 5, 10, 15 mg/L). (e) Turbidity removal at 1 mg/L alum and 1 mg/L longan seed powder with varying pH (4, 5, 6, 7, 8 and 9). (f) Turbidity removal at 1 mg/L alum and pH 4 with varying concentration of longan seed powder (0, 0.25, 0.5, 1, 2.5 and 5 mg/L).

200 NTU was treated with a constant quantity of KAl within the pH range of 4–9. Based on previous studies suggesting an optimal alum dosage of 10–40 mg/L [55], we selected 10 mg/L for this initial experiment. As shown in Fig 4a, a pH of 4 resulted in the highest percentage of turbidity removal across all initial turbidity levels. Specifically, turbidity removal percentages were 97.2 ± 1.1%, 93.4 ± 1.1%, 78.4 ± 2.5%, 48.0 ± 3.3%, and 25.0 ± 1.6% from initial turbidity levels of 200, 100, 50, 25, and 10 NTU, respectively.

Based on these findings, subsequent experiments were conducted at pH 4 to determine the optimal KAl concentration, as illustrated in Fig 4b. Synthetic water samples with turbidity levels of 10–200 NTU were treated with varying KAl concentrations (0, 1, 2.5, 5, 10, and 15 mg/L). KAl concentrations of 5, 10, and 15 mg/L demonstrated significant turbidity removal, with 5 mg/L generally being the most effective. Notably, at an initial turbidity of 25 NTU, the optimal KAl concentration was 10 mg/L. The turbidity removal percentages recorded were 96.5 ± 1.9%, 92.6 ± 2.0%, 79.7 ± 1.7%, 57.0 ± 2.2%, and 27.3 ± 3.2% for initial turbidity levels of 200, 100, 50, 25, and 10 NTU, respectively.

The results indicate that pH significantly influences the efficacy of the coagulant (KAl). Turbidity removal increases as pH decreases, primarily due to two factors. First, the coagulant acquires a higher positive charge in acidic media, enhancing its attraction to negatively charged particles in the water. Second, a lower pH reduces hydrogen bonding sites on the coagulant, facilitating more effective bridge bond formation and charge neutralization [56].

The results reflect the combined effects of coagulation and flocculation processes. The initial turbidity level dose influences performance. As initial turbidity values increase from 10 to 25 NTU, a corresponding decline in performance is observed. An increase in turbidity correlates with enhanced performance in turbid water with NTU values of 50, 100, and 200. The formation of floccules can occur via charge neutralization. In elevated raw water turbidity conditions, the coagulant effectively activates its capacity to neutralize the inherent negative charges of particles in the water. Consequently, reduced levels of final water turbidity are noted [57]. All data are presented as mean ± SD (n = 3); statistical comparisons (ANOVA, p < 0.05) confirmed significant differences among doses at each turbidity level.

## pH and dose optimum of longan seed powder as a coagulant in synthetic turbid water

The performance of longan seed powder as a coagulant for turbid water treatment was evaluated following the same procedure as for alum. We first determined the optimal pH, followed by the optimal concentration. As shown in Fig 4c, the pH levels of synthetic turbid water samples were adjusted to 4, 5, 6, 7, 8, and 9, and treated with 10 mg/L longan seed powder. However, turbidity reduction did not exceed 25% at any pH level. Subsequently, we investigated the optimal longan seed powder concentration (0, 1, 2.5, 5, 10, and 15 mg/L) at pH mirroring the optimal pH for alum in Fig 4d.

Again, turbidity reduction remained below 25%. This may be attributed to the influence of pH on starch characteristics. While low pH levels reportedly increase starch crystallinity, it also diminishes starch processability and can lead to hydrolysis at pH 4 [58]. These results support using LSP synergistically with alum rather than as a standalone primary coagulant under the tested conditions.

## pH and dose optimum of longan seed powder as a coagulant aid with alum in synthetic turbid water

To investigate the effects of pH on turbidity removal of longan seed powder as a coagulant aid to alum synthetic turbid water samples (10, 25, 50, 100 and 200 NTU) were adjusted to pH levels ranging from 4 to 9. Longan seed powder (1 mg/L) and alum (1 mg/L) were added into these turbid water samples. Alum concentration of 1 mg/L was chosen based on preliminary observation showing suboptimal turbidity removal (<20%), suggesting the need for a coagulant aid. As natural polysaccharide (starch-based) doses typically range from 0.5 to 2.5 mg/L [59], we selected 1 mg/L longan seed powder for optimal pH determination. Fig 4e indicates that a pH of 4 consistently demonstrated the highest turbidity removal performance across all initial turbidity levels, specifically 91.9 ± 2.1%, 85.4 ± 6.4%, 80.8 ± 3.9%, 45.4 ± 5.6%, and 18.4 ± 0.8% for 200, 100, 50, 25, and 10 NTU, respectively. Subsequently, we conducted experiments at pH 4 and 1 mg/L alum to determine the optimal concentration of longan seed powder as coagulant aid. Fig 4f illustrates the turbidity

removal efficiency for synthetic water samples (10–200 NTU), treated with varying longan seed powder concentration at 0, 0.25, 0.5, 1, 2.5, and 5 mg/L. Longan seed powder concentrations of 0.5, 1, and 2.5 mg/L demonstrated significant turbidity removal, with 0.5 mg/L identified as the optimal concentration, achieving 96.7±1.2%, 91.3±0.9%, 85.0±4.1%, 49.7±2.9%, and 25.9±2.8% for initial turbidity levels of 200, 100, 50, 25, and 10 NTU, respectively.

Longan seed powder demonstrates as an optimal coagulant aid properties at pH of 4, a KAl concentration of 1 mg/L, and a dosage of 0.5 mg/L longan seed powder. The effectiveness of starch-based coagulant aid is supported by previous studies. For instance, a combination of PAC coagulant and rice starch coagulant aid achieved turbidity removal exceeding 96.6% under optimal conditions (pH 6, coagulant dose 7.5 mg/L, coagulant aid dose 0.025 mg/L, and turbidity 37.5 NTU) [60]. Similarly, poly-aluminium chloride (5 mg/L) combined with corn starch (0.7 mg/L) reduced turbidity by 98.48% from an initial level of 250 NTU [61]. The dose-response plots

### pH and dose optimum of longan seed powder as a coagulant aid with Alum in raw water

Raw water, exhibiting an initial turbidity of 50 NTU, was collected from the Kok River in the Mueang Chiang Rai District of Chiang Rai, Thailand, at coordinates 19°55'29.1"N 99°51'56.5" E. As shown in Fig 5a, pH suitability was assessed within the 4–9 range, a 10 mg/L KAl concentration. The optimal turbidity removal percentage (79.8±1.7%) was achieved at pH 4. At this pH, the optimum concentration of Alum for turbidity reduction was determined to be 5 mg/L, resulting in an 82.0±2.4% reduction in turbidity (Fig 5b).

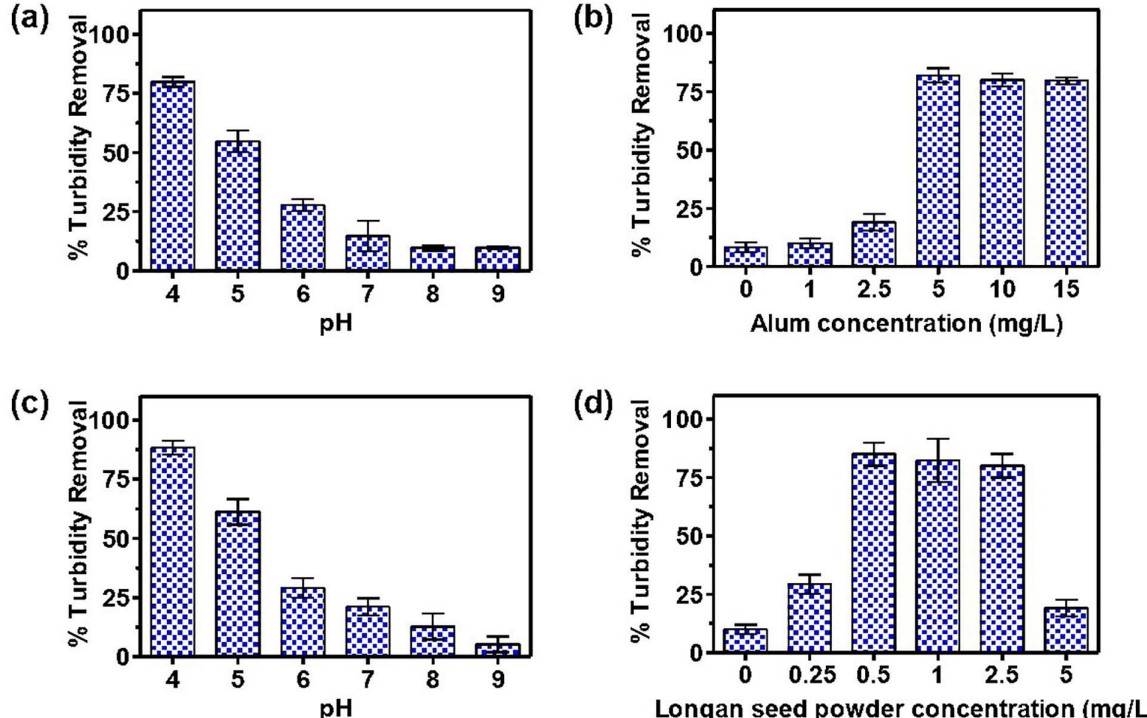

**Fig 5. Effect of pH and Alum as coagulant and longan seed powder as coagulant aid dosage on the efficiency of turbidity removal in raw water (approximately 51 NTU).** (a) Turbidity removal with 10 mg/L alum at varying pH (4, 5, 6, 7, 8 and 9) levels. (b) Turbidity removal at pH4 with varying alum concentrations (0, 1, 2.5, 5, 10 and 15 mg/L). (c) Turbidity removal with 1 mg/L alum and 1 mg/L longan seed powder at varying pH (4, 5, 6, 7, 8 and 9) levels. (d) Turbidity removal at pH 4 with 1 mg/L alum and varying longan seed powder concentrations (0, 0.25, 0.5, 1, 2.5 and 5 mg/L).

Fig 5c illustrates the experiment to determine the optimal concentration of longan seed powder, used in conjunction with 1 mg/L of Alum, for turbidity removal. The highest turbidity removal percentage (88.3±2.3%) was achieved at pH 4. Meanwhile, Fig 5d shows the effect of longan seed powder concentration in turbidity removal for raw water (approximately 51 NTU) at pH 4, in conjunction with 1 mg/L of Alum. The turbidity removal percentages were significantly higher with longan seed concentrations of 0.5, 1.0, and 2.5 mg/L at a turbidity reduction of 85.0±4.1%, 82.3±1.0%, and 80.0±4.1%, respectively. Therefore, the optimal conditions for using longan seed powder as a coagulant aid in raw water are: pH 4, 1 mg/L KAI, and 0.5 mg/L longan seed powder. Notably, the addition of 0.5 mg/L longan seed powder enabled a fivefold reduction in alum concentration (from 5 mg/L to 1 mg/L) while maintaining effective turbidity removal.

## The effect of Alum and longan seed powder on pH and TDS

Figs 6a and 6c show the pH and TDS changes according to the experiment in Fig 5b (alum only), while Figs 6b and 6d correspond to the experiment in Fig 5d (alum and longan seed powder). The optimum alum concentration at pH 4 was determined. The initial pH of the experiment was established at 4. Alum addition decreases the water pH. However, the pH change was less significant with the combined addition of alum and longan seed powder. This less significant pH change might be attributed to the lower alum concentration used in conjunction with longan seed powder, as KAI exhibited optimal performance at reduced dosages in acidic conditions. Based on previously reported data, similar changes were observed when Alum and polyaluminium chloride were utilized as coagulants in wastewater treatment. Upon adding Alum

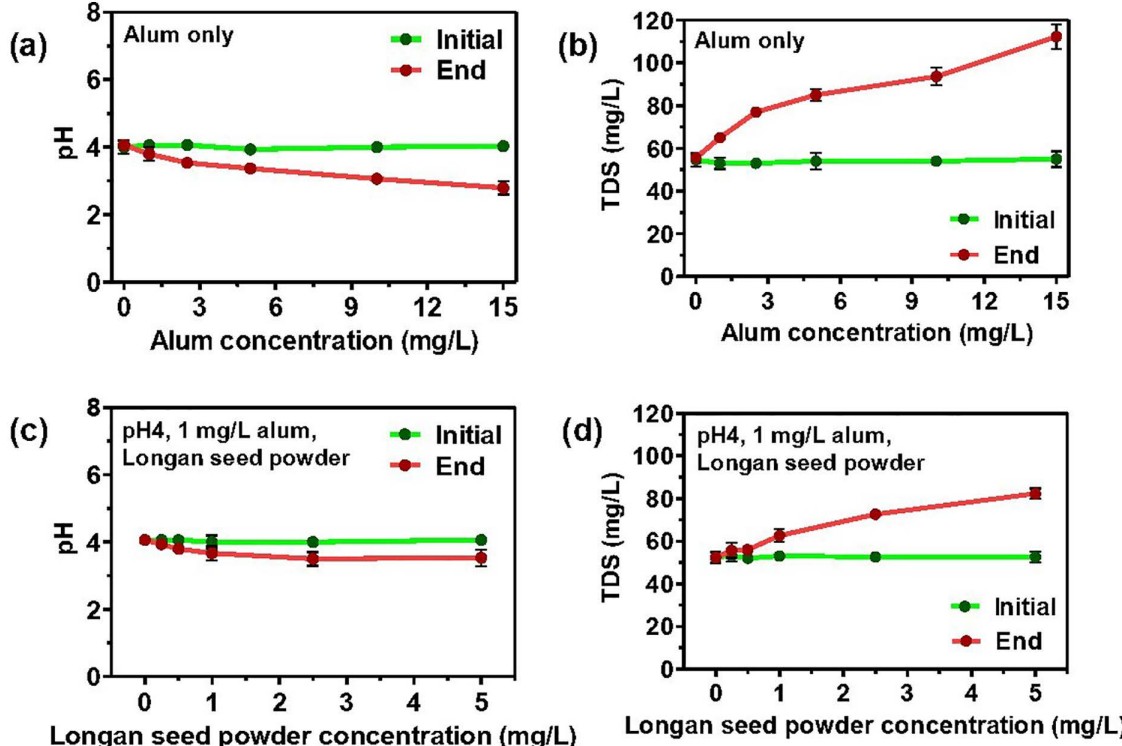

**Fig 6. Effect of addition of alum and longan seed powder on the pH and TDS of raw water.** Changes in a) pH and b) TDS with the addition of alum observed in experiment in Fig 5b, and Changes in c) pH, d) TDS with the addition of 1 mg/L alum and longan seed powder observed in experiment in Fig 5d.

to water, it undergoes hydrolysis, resulting in the formation of aluminum hydroxide and a decrease in the pH of the water due to the generation of acidic species [62, 63].

The raw water sample used in this study has an initial total dissolved solids (TDS) of 50 mg/L, a contribution from the solution used in pH adjustment. Fig 6b shows that the TDS level increases with the addition of alum, indicating the dissolution of alum salt. Figs 5c and 5d illustrate that it is possible to maintain high turbidity removal with the combination of 1 mg/L alum and 0.5–2.5 mg/L longan seed powder without significantly impacting the pH and TDS levels.

**Coagulation-flocculation Process by Alum**

The addition of alum (aluminum sulfate) at low concentrations (1 and 2.5 mg/L) does not effectively precipitate particles, even in turbid water. This can be attributed to several factors. Firstly, alum and coagulation. Alum is commonly used in water treatment to coagulate and precipitate suspended solids. It works by neutralizing the charges on colloidal particles, which facilitates their aggregation and subsequent removal from the water through sedimentation. Secondly, insufficient

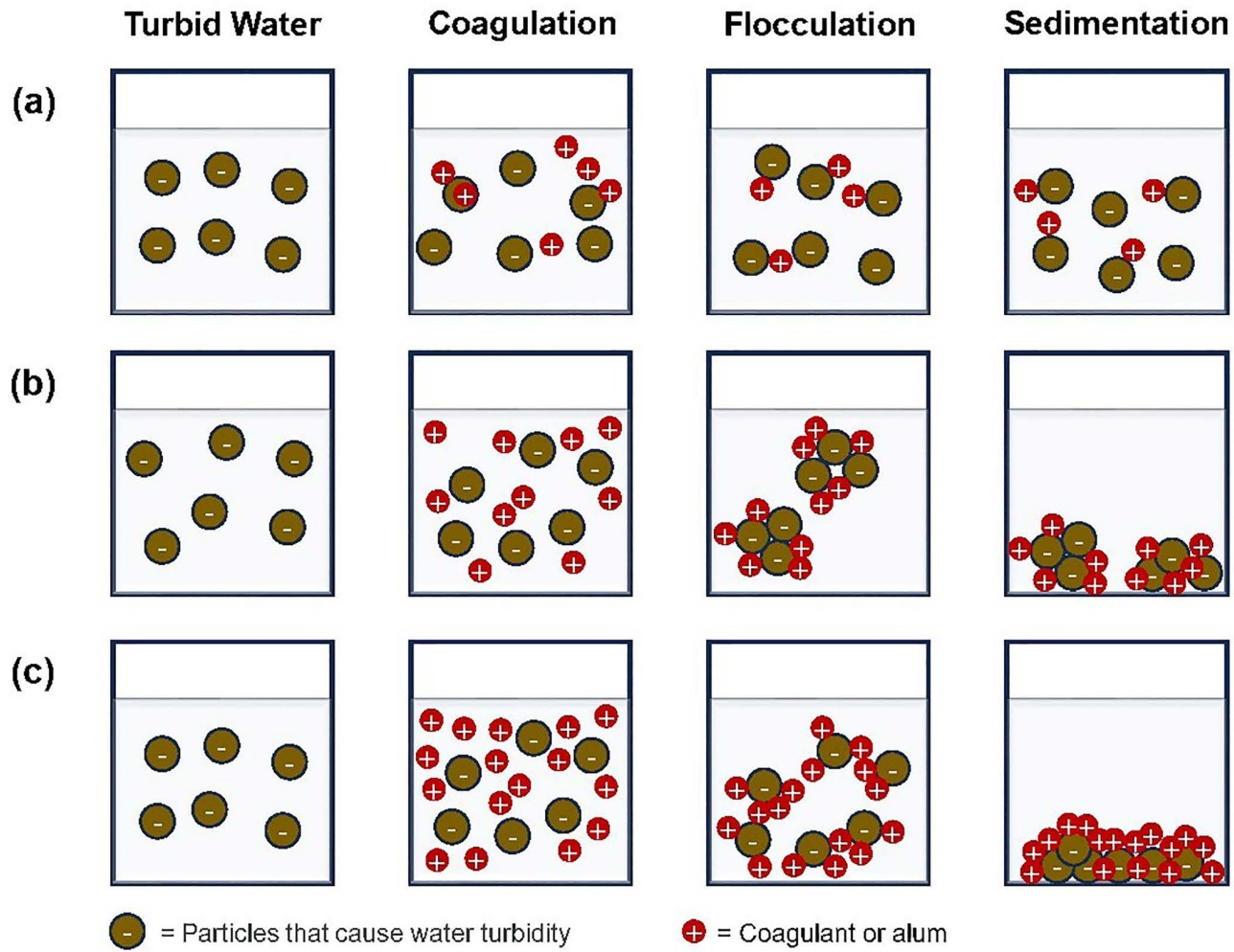

**Fig 7. Illustration of coagulation-flocculation mechanisms with low concentration of alum (a); optimal concentration of alum (b); and excessive concentration of alum (c).**

concentration. At low concentrations (1 and 2.5 mg/L), the amount of alum may be insufficient to fully neutralize the charges on the colloidal particles, leading to only partial coagulation. As shown in Fig 7a, turbidity remains high at these dosages. Thirdly, salting-in effect. Paradoxically, the presence of ions from alum can sometimes increase the solubility of certain compounds. This "salting-in" effect may stabilize colloidal particles at low alum concentrations, preventing their sedimentation. Finally, colloidal kaolin and turbidity. Kaolin clay, often responsible for turbidity, has negatively charged particles. With insufficient alum, these particles cannot aggregate effectively, and the water remains turbid. To achieve efficient precipitation and improve water clarity, higher alum concentrations may be necessary [64].

When Alum is added at the right concentration (Fig 7b), alum destabilizes and coagulates suspended particles through adsorption and charge neutralization. This involves neutralizing the particle charge, reducing or completely removing electrostatic repulsion, promoting agglomeration. With no net charge on the particle surface, the formation of an electrical double layer is stopped, and particle aggregation is driven by van der Waals forces [65]. Charge neutralization is achieved through a systematic 'patchwise' approach. Adsorption occurs between the high-charge density electrolytes (coagulants) and low-charge density colloidal particles. The electrostatic patch mechanism shows distinct positive and negative regions on a particle's surface [66]. When the charge density increases, less coagulant is needed for water treatment [67]. Excess coagulant application may reverse and redistribute the colloid's charge, resulting in a positive colloid or re-stabilization.

As illustrated in Fig 7c, excessive alum application leads to sweep floc. This involves the formation of an insoluble precipitate (amorphous metal hydroxide) that encapsulates colloidal particles, effectively removing them from the water. This non selective mechanism, also known as sweep coagulation, is particularly effective for destabilizing turbid water, especially with low turbidity, as it increases the probability of colloidal interaction with water by adding many precipitate particles [68, 69]. Sweep floc occurs predominantly in water treatment applications when the pH is maintained between 6 and 8 (neutral), and coagulant salts (Al or Fe) are employed at concentrations relevant to the generated amorphous metal hydroxide solid, generally exceeding adsorption levels. This results in increased chemical expenses and denser, harder-to-dewater sludge [69]. However, sweep coagulation offers a distinct benefit in process control compared to destabilization by adsorption and charge neutralization, as it is less sensitive to dosage variations, particularly overdosing. Factors influencing sweep coagulation include oversaturation (requiring a higher precipitate concentration for rapid precipitation), the presence of anions (especially sulfate, which enhances precipitation rate), and colloid concentration (higher colloid content provides more nuclei for precipitate formation) [70–73].

## Coagulation-flocculation process by alum and longan seed powder

In this mechanism, polysaccharides with highly reactive surfaces and linear or branched structures act as coagulant aids, promoting inter-particle bridging and enhancing micro-floc aggregation during flocculation [31]. As shown in Fig 8a, insufficient polymer addition can result in increased turbidity. This may be because the polysaccharides bind to only one particle or fail to adequately aggregate more particles into larger flocs. Polymer chains adhere to particle surfaces through (i) coulombic interactions, (ii) dipole interactions, (iii) hydrogen bonding, and (iv) van der Waals forces of attraction. The remaining polymer chain protrudes into the water and binds to other particle surfaces, forming bridges and larger, more readily settleable flocs (Fig 8b).

Furthermore, adequate unoccupied surface area on a particle is crucial for polymer bridging, as it allows for the attachment of polymer chain segments adsorbed on other particles [74]. In cases where excess polymer is present or no further particles are available (Fig 8c), the original particle can be fully enveloped by the free extended segments of the polymer molecule, thus restabilizing the colloid. In addition to the factors mentioned previously, polymer bridging is also influenced by the molecular weight of the polymer, the charge density of polyelectrolytes, and ionic strength of colloid particles. A study has summarized these effects, indicating that: (i) linear, high-molecular-weight polymers are the most effective; (ii) minimal adsorbed polymer is necessary, as excessive amounts can lead to restabilization; (iii) optimal charge density benefits polyelectrolytes; (iv) specific metal ions and ionic strength can influence outcomes; and (v) while the bridging process

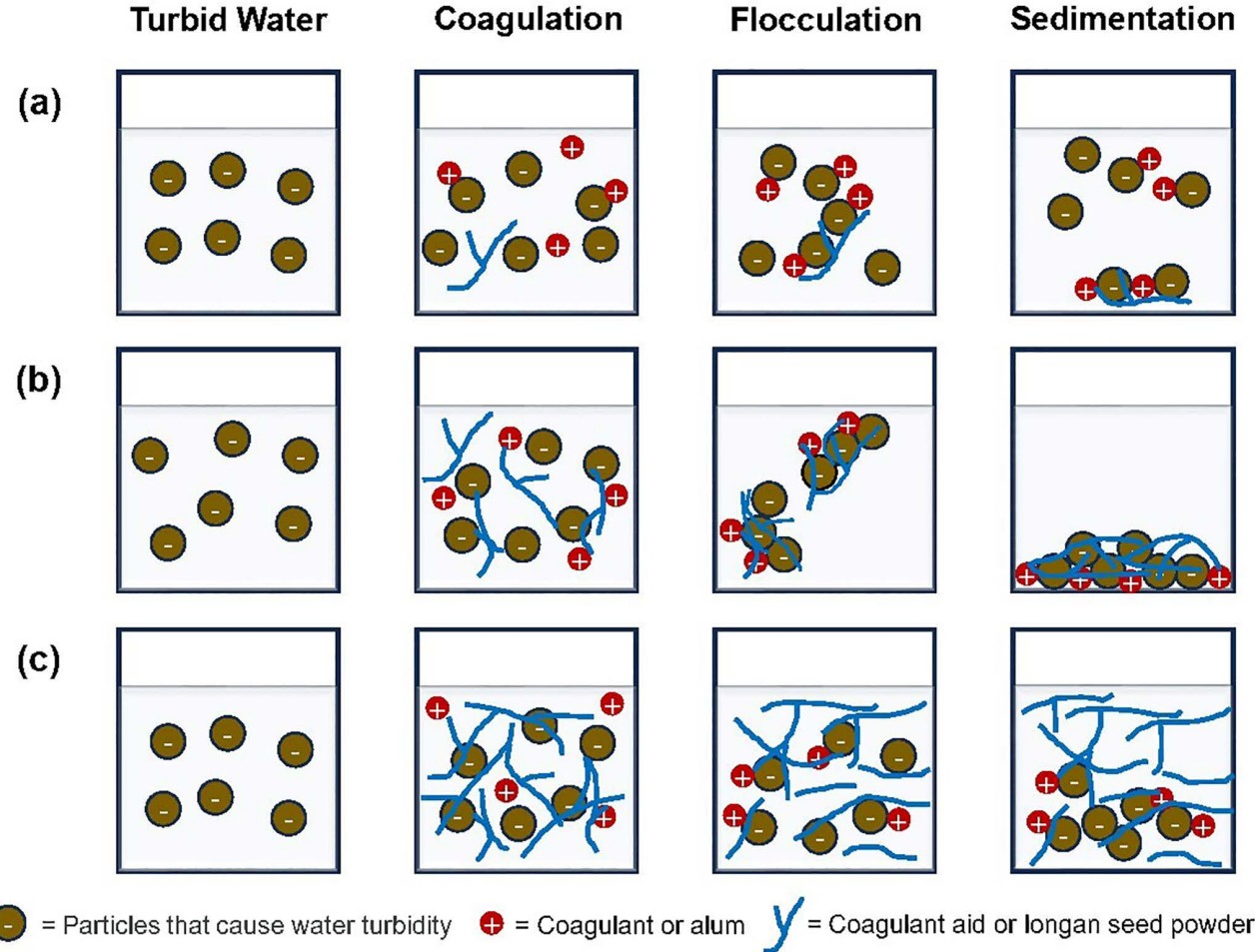

**Fig 8. Illustration of coagulation-flocculation mechanisms used concentration of Alum and initial turbidity remained constant with different concentrations of longan seed powders (a) Low concentration longan seed powders, (b) longan seed powders with the appropriate concentration and (c) longan seed powders with excessive concentration.**

produces robust flocs, damaged flocs may be irreversible [74]. Together with FT-IR/SEM evidence (functional groups for adsorption/bridging; particle size 2–20 μm), these results integrate characterization with performance to support the proposed dual mechanism.

To address the reviewer's request, we added Table 4 comparing our optimized LSP + alum system with representative literature on plant-based coagulants. For example, PAC + rice starch achieved >96.6% removal at pH 6 with 7.5 mg/L coagulant and 0.025 mg/L aid (37.5 NTU) [60], and PACl (5 mg/L) + corn starch (0.7 mg/L) achieved 98.48% at 250 NTU [61]. In our study, alum (1 mg/L) + LSP (0.5 mg/L) achieved up to 96.7% removal (200 NTU) at pH 4, while reducing alum dosage fivefold compared with alum-only optimum in raw water.

## Cost analysis

The cost analysis of producing longan seed powder (LSP) as a bio-coagulant demonstrated strong economic potential, particularly when utilizing agricultural waste as raw material. The experimental results indicated that the yield of LSP after drying and grinding was approximately 39.23% of the original raw seed weight. This corresponds to a conversion rate

whereby 1 kg of raw longan seeds produces approximately 0.3923 kg of LSP. Consequently, the production of 1 kg of LSP requires approximately 2.55 kg of raw seeds.

The production process comprised several sequential steps: cleaning, drying in a hot-air oven at 60°C for 24 hours, grinding, and sieving. The total electricity cost per kilogram of raw seeds was estimated at THB 1.712, while labor and maintenance contributed THB 0.208. Miscellaneous costs, including depreciation, packaging, logistics, indirect labor, and utilities, were calculated at THB 1.916. As the raw longan seeds are classified as agricultural waste, the material cost was considered zero. Thus, the total production cost per kilogram of raw seeds amounted to THB 3.836 (USD 0.12).

Based on the required input mass of 2.55 kg of raw seeds to produce 1 kg of LSP, the laboratory-scale production cost was determined to be THB 9.78 (USD 0.30) per kilogram. This value was derived by multiplying the unit cost per kilogram of raw seeds by the corresponding input mass, as summarized in Table 3. Notably, this cost is significantly lower than the market price of conventional coagulants, such as aluminum ammonium sulfate, which is priced approximately THB 13 (USD 0.40) per kilogram in Thailand. Although this comparison does not account for additional operational expenses such as logistics and labor at industrial-scale LSP production, the use of LSP could reduce overall treatment costs, as the consumption of alum can be reduced up to fivefold when LSP is applied as a coagulant aid. Furthermore, the production cost of LSP is expected to decrease substantially at industrial scale benefiting from economies of scale and process optimization.

Thailand's longan industry generates a considerable quantity of seed byproduct annually, providing a consistent and sustainable feedstock for LSP production. This availability underscores the potential of LSP as a cost-effective and environmentally sustainable alternative to conventional coagulants, supporting circular economy and waste valorization initiatives.

The use of longan seed powder presents not only a cost-effective alternative but also an environmentally friendly solution. Similar to other plant-based coagulants like Moringa oleifera, longan seed powder contributes to the reduction of chemical use in water treatment and minimizes harmful sludge generation [75, 76]. It also encourages the use of renewable agricultural residues, aligning with principles of circular economy and sustainable resource management. Okuda et al. (2001) [76] also demonstrated that plant-based coagulants are effective due to their protein-based active compounds, which promote particle aggregation efficiently.

These findings support the notion that integrating longan seed powder into coagulation processes could be both economically and ecologically advantageous. This approach holds particular promise for decentralized or small-scale water treatment systems, where resource constraints make low-cost, sustainable options especially valuable [76].

### Eco-friendliness, stability, and practicality

The eco-friendly claim is supported by lower alum usage (fivefold reduction in raw water), which implies lower sludge generation and reduced chemical inputs. While toxicity and biodegradability tests were beyond the present scope, future work will evaluate LSP biodegradation in soil/water, acute/chronic toxicity, and sludge dewatering behavior to quantify environmental benefits.

**Table 3. Summary of production cost of longan seed powder (per 1 kg) produced from 2.55 kg of longan seed.**

| Cost Component | Cost (THB) | Percentage (%) |
|---|---|---|
| Raw material (longan seeds) | 0.00 | 0.00 |
| Electricity (drying and grinding) | 4.37 | 44.64 |
| Labor and maintenance | 0.53 | 5.41 |
| Miscellaneous (utilities, packaging, etc.) | 4.89 | 49.95 |
| **Total Cost** | **9.78** | **100** |

Storage stability of plant extracts/powders is practically important; we recommend assessing moisture-controlled storage (≤40% RH, 25 °C), shelf-life over 3–6 months, and potency retention under field conditions in subsequent studies.

## Conclusion

This study demonstrates that longan seed powder (LSP), an abundant agricultural byproduct in northern Thailand, possesses significant potential as a sustainable and low-cost coagulant aid in water purification. When used alone, LSP achieved limited coagulation efficiency (<25% turbidity removal). However, when combined with 1 mg/L alum, at an optimized dosage of 0.5 mg/L LSP and pH 4, it achieved 96.7 ± 1.2% turbidity removal, comparable to that of alum alone at 5 mg/L (96.5 ± 1.9%). This indicates a fivefold reduction in alum dosage without compromising treatment efficiency. The process also minimized changes in pH and total dissolved solids (TDS), suggesting that longan seed starch plays a synergistic role in charge neutralization and polymer bridging, as confirmed by FT-IR and SEM analyses showing typical polysaccharide functional groups and compact surface morphology conducive to adsorption and bridging mechanisms.

Economically, the cost of laboratory-scale production of longan seed powder was calculated at THB 9.78/kg (USD 0.30/kg), markedly lower than that of commercial alum, approx. THB 13/kg (USD 0.40/kg). Environmentally, replacing alum with longan seed powder could potentially reduce sludge formation and aluminum residues, improving post-treatment sludge management and minimizing ecological toxicity. The use of such natural polymers aligns with circular economy principles by valorizing agricultural waste, thereby supporting local sustainability and community-based water management initiatives.

Nevertheless, this study has certain limitations. The experiments were conducted under controlled laboratory conditions using both synthetic and raw river water. Variability in the composition of longan seeds (starch and polysaccharide ratios), as well as differing environmental parameters (temperature, ionic strength, and organic matter), may influence coagulation performance under field conditions. Further research should therefore focus on (i) evaluating LSP stability and shelf-life during storage, (ii) assessing its biodegradability and potential ecotoxicity, (iii) testing scalability and performance across different water types (e.g., domestic, industrial, and surface waters), and (iv) optimizing extraction or partial purification methods to identify active biopolymers responsible for coagulation.

In summary, this study fills a critical knowledge gap by systematically evaluating longan seed powder as a coagulant aid for both synthetic and raw river water. It demonstrates the feasibility of reducing chemical coagulant use by up to 80%, achieving high removal efficiencies, and reducing treatment costs. The outcomes not only contribute to advancing sustainable water treatment technologies but also establish a foundation for future pilot-scale or decentralized implementations that integrate economic, environmental, and social benefits.

## Author contributions

**Conceptualization:** Anuwat Aunkham, Vivat Keawdounglek.

**Data curation:** Anuwat Aunkham, Wei Chung Sim.

**Formal analysis:** Anuwat Aunkham, Wei Chung Sim.

**Funding acquisition:** Anuwat Aunkham.

**Investigation:** Anuwat Aunkham.

**Methodology:** Anuwat Aunkham, Vivat Keawdounglek, Budsakorn Aiyee, Ploypairin Choochan, Phitchayapha Chumueang, Wichuda Intawong, Natcha Thambun, Anantaya Anantaburiand, Panadda Klinchan.

**Project administration:** Anuwat Aunkham.

**Resources:** Anuwat Aunkham.

**Software:** Anuwat Aunkham.

**Supervision:** Anuwat Aunkham, Vivat Keawdounglek.

**Validation:** Anuwat Aunkham.

**Visualization:** Anuwat Aunkham.

**Writing – original draft:** Anuwat Aunkham.

**Writing – review & editing:** Anuwat Aunkham, Wei Chung Sim.

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
