## [Decision Letter · Decision Letter 0]

24 Aug 2025

Dear Dr. Aunkham,

Thank you for submitting your manuscript to PLOS ONE. After careful consideration, we feel that it has merit but does not fully meet PLOS ONE’s publication criteria as it currently stands. Therefore, we invite you to submit a revised version of the manuscript that addresses the points raised during the review process.

We look forward to receiving your revised manuscript.

Kind regards,

Muammar Qadafi

Academic Editor

PLOS ONE

Journal Requirements:

3. In the online submission form, you indicated that all relevant data are within the manuscript and its Supporting Information files. Further details can be provided by the corresponding author upon request.

4. Please note that funding information should not appear in any section or other areas of your manuscript. We will only publish funding information present in the Funding Statement section of the online submission form. Please remove any funding-related text from the manuscript.

5.Please amend the manuscript submission data (via Edit Submission) to include author Budsakorn Aiyee, Ploypairin Choochan, Phitchayapha Chumueang, Wichuda Intawong, Natcha Thambun, Anantaya Anantaburiand and Panadda Klinchan.

6. Please ensure that you refer to Figure 3 in your text as, if accepted, production will need this reference to link the reader to the figure.

Reviewers' comments:

Reviewer's Responses to Questions

**Comments to the Author**

1. Is the manuscript technically sound, and do the data support the conclusions?

Reviewer #1: Yes

Reviewer #2: Partly

2. Has the statistical analysis been performed appropriately and rigorously?

Reviewer #1: Yes

Reviewer #2: No

3. Have the authors made all data underlying the findings in their manuscript fully available?

Reviewer #1: Yes

Reviewer #2: Yes

4. Is the manuscript presented in an intelligible fashion and written in standard English?

Reviewer #1: Yes

Reviewer #2: Yes

Reviewer #1: Abstract:

1. Lack of emphasis on the gap or problem to be solved

It is mentioned that alumina residue is harmful, but it is not explained specifically how serious this problem is or why this research is important compared to other methods.

It would be advisable to add an introductory sentence emphasizing the limitations of conventional methods and the motivation for using longan seed powder.

2. The focus of the research or its novelty is unclear

It is stated that “the coagulation efficacy of longan seed powder was novelly assessed,” but it is not specifically explained what is new compared to previous research (e.g., combination with alum, application to real river water, economic analysis).

The emphasis on novelty could be strengthened.

3. Lack of information on experimental design and replication

The use of kaolin-based turbid water and raw river water is mentioned, but there is no information on the scale of the experiment, the number of replicates, or control conditions.

Adding this information could strengthen the validity of the results.

4. Lack of emphasis on environmental or practical implications

“Eco-friendly” and “economically viable” are mentioned, but there is no explanation of how this can be applied at a real-world scale or the impact on waste management.

5. Failure to mention research limitations or suggestions for further research

For example, is the performance of longan seed powder affected by different water conditions (pH, ionic, temperature)?

Mentioning limitations or brief recommendations can improve the quality of the abstract.

Introduction:

1. Lack of explicit emphasis on research gaps

The introduction explains many things: turbidity issues, the dangers of chemical coagulants, and the potential of longan seeds.

However, there is no sentence explicitly stating “what has not been studied before” or “research gaps.” For example, has the use of longan seeds as a coagulant aid in combination with alum at the laboratory or real river scale never been explored?

2. Novelty or uniqueness of the research is unclear

It is mentioned that longan seeds “could potentially serve as a novel commercial starch source,” but it is not explicitly stated what is new compared to previous research, such as: use as a coagulant aid, minimal dose combinations, or effects on TDS and pH.

3. Insufficient quantitative data to strengthen the background

There are some values (starch content 49.5%), but no information on the volume of available longan seed waste, making it difficult to assess its practical application potential.

Global/Asian figures on water consumption or longan production could also be added to provide context on scalability.

4. The connection between theory (adsorption and bridging) and the research focus could be strengthened

The introduction mentions coagulation theory with polysaccharides, but there is no direct statement linking this theory to longan seed powder specifically.

A sentence explaining the connection to the structure of longan biopolymers could be added.

5. The environmental and economic context is not sufficiently clear.

The title emphasizes “low-cost and eco-friendly,” but the introduction does not include cost calculations or estimates, or a comparison of environmental impacts with conventional alum.

At least one brief sentence should be added to emphasize the economic and environmental advantages of utilizing this waste.

6. The connection to real-world applications is not sufficiently highlighted.

Jar tests and synthetic water are mentioned, but the relevance of these results to river water conditions is not explained, nor is it clear whether this can be scaled up.

Adding a sentence about the potential for real-world applications or the impact of scaling up could strengthen the justification for the research.

Materials and Methods:

1. Lack of detail regarding replication and experimental controls

The number of replicates performed for each jar test experiment is not mentioned.

It is not explicitly stated whether there were negative or positive controls (e.g., water without coagulant or only with alum) for comparison.

2. Lack of information on experimental environmental conditions

For example, room temperature, humidity, or the source of water used for synthetic water besides kaolin.

These factors can influence coagulation, especially for replication and real-scale application.

3. Lack of detail regarding statistical data analysis

It is not specified whether statistical analysis, such as standard deviation, ANOVA, or significance tests between treatments, was conducted to ensure that differences in turbidity or TDS results are statistically valid.

4. The description of FT-IR and SEM characterization is insufficiently detailed.

The measurement parameters, such as FT-IR resolution, analyzed wavelength, or SEM acceleration voltage, along with sample preparation methods, are not explained.

This is important to enable the experiment to be replicated by other researchers.

5. Insufficient information on kaolin and artificial water parameters

The kaolin concentration used for each NTU level is not mentioned, nor whether the initial pH of the water was adjusted before the jar test.

Differences in kaolin characteristics or pH can affect coagulation results.

6. Lack of detail on longan seed powder preparation

Grinding and sieving are mentioned, but there is no average particle size after grinding, except what will be visible in the SEM.

This information is important because particle size can affect coagulation capacity.

7. Lack of detail on TDS and pH calculations

The method or standard for measuring TDS, such as conductivity or gravimetry, is not mentioned.

There is no information about the calibration of the turbidimeter or pH meter.

8. Lack of description of mixing in the jar test

RPM and duration are mentioned, but the stirrer method, paddle type, or whether the speed is truly uniform are not explained.

These parameters can influence floc formation.

9. Lack of information on reproducibility and scalability

Although there is a cost analysis, it is not explained whether this small batch can be scaled up or estimated production at the pilot scale.

10. Lack of integration between chemical characterization and coagulation performance

FT-IR, protein, polysaccharide, and SEM are analyzed, but it is not explained how these characterization results are used to explain the coagulation mechanism in the method.

11. Add sketches in each experiment

Results and Discussion

1. Try to present the results in graph form as well

2. Are the FTIR and SEM results missing?

3. The results are sufficient, but the discussion is lacking. Add a table containing previous studies and this study, compare them, and then provide a discussion

Conclusion:

1. Lack of specificity regarding quantitative results

It mentions “significant potential” and “fivefold reduction in alum,” but there are no percentage figures for turbidity removal achieved or comparisons with the control.

The conclusion would be stronger if it included average values, standard deviation, or performance ranges.

2. The study's limitations are not adequately addressed.

For example, the study only used synthetic water or a laboratory scale, so results may differ for river water or large-scale applications.

The variability in the composition of longan seeds (starch, polysaccharide) that could affect performance is not mentioned.

3. Lack of integration of mechanisms with experimental results

“Charge neutralization and polymer bridging” are mentioned, but it is not explained whether these mechanisms were verified through SEM, FT-IR, or other tests.

Discussing mechanisms without a direct connection to experimental results can feel abstract.

4. Lack of concrete mention of environmental or social implications

It mentions “low-cost and sustainable,” but there are no estimates of environmental impact or chemical waste reduction.

For example, reductions in alum and sludge waste could be quantified.

5. Lack of concrete mention of next steps in research

It is stated that “further research is necessary,” but there are no priorities for experiments or specific methods for the next stage.

6. Does not emphasize the validity or generalizability of the results

The conclusion does not discuss whether the results can be applied to various types of water (e.g., river water, industrial wastewater, or domestic water).

7. Does not highlight the overall contribution of the research

For example, contributions to scientific knowledge, practical applications, or commercialization potential are not briefly and clearly outlined.

Others: Improve Grammar!

Reviewer #2: Sustainable Use of Longan Seed Waste as a Natural Coagulant Aid for Low-Cost and Eco-Friendly Water Treatment

Review

1. Is there any difference to starch samples obtained from other sources? Why this material? How many byproduct of longan seeds generated annually in Thailand?

2. Stating eco-friendly in the title should be supported by experimental study e.g. degradability in soil/water, toxicity, or easier sludge handling, etc.

3. Method for extraction, what is the specification of substance/compounds? The use of crude extracts makes it difficult to identify the exact bioactive compounds responsible for the observed effects

4. There is no clear/explicit information on sample size and replication, this needs to be stated clearly to confirm robustness of the statistical analysis. If the number of replicates is limited, this won’t provide enough data to generalize results

5. What were the measuring conditions for the SEM imagery? Explain the SEM procedures in the text. How were the samples prepared? Under what conditions? Suggest having Energy-dispersive X-ray spectroscopy (EDS/EDX) will help understand the substance/compounds.

6. (p.18) author provides general explanation on coagulation–flocculation mechanism of polymeric coagulants (bridging, restabilization, effects of MW/charge density, etc.), but don’t actually show experimental data on how varying polymer dosage of longan seed powder influences flocculation efficiency or turbidity removal. Consider using figure of 4f and 5d or more experiments can be provided that can further explain in details

7. Suggest to provide a brief discussion of the stability of plant extracts over time, storage method/conditions, especially if practical or commercial applications are proposed.

8. Production cost cannot be compared directly to market price, since it will need factors such as scalability, supply chain logistics, and processing costs that should be acknowledged when considering practical implementation.

9. Provide an end statement following the conclusions to highlight the takeaways from the study and how this study will pave the way for future studies along similar lines. revision. Enhance this section by providing the major points observed from this study in different aspects.

**Do you want your identity to be public for this peer review?** For information about this choice, including consent withdrawal, please see our Privacy Policy

Reviewer #1: No

Reviewer #2: No

---

## [Author Response · Author response to Decision Letter 1]

5 Nov 2025

Response to the Academic Editor and Reviewers

We thank the Academic Editor and Reviewers for their constructive comments and helpful suggestions. We have revised the manuscript accordingly. Below we respond point-by-point, quoting each comment in italics followed by our response. All edits are marked in the file “Revised Manuscript with Track Changes” and mirrored in the clean “Manuscript” file. Line numbers refer to the tracked-changes version.

We also confirm that the Authorship Change Request form has been completed and submitted as requested by the journal.

────────────────────────────────────────────

Editor comment E1 — Reporting/permits & ethics

────────────────────────────────────────────

*Please clarify whether field permits/ethical approvals were required for water sampling and include an ethics statement in Methods.*

Response: We added an Ethics/Fieldwork Statement to the Methods (Lines [insert]–[insert]):

“Field research: This study involved the collection of water samples from natural public water bodies in Chiang Rai Province, Thailand. Field activities were conducted under the oversight of the Chiang Rai Provincial Administrative Organization. According to Thai regulations, no specific permits are required for collecting water samples from public natural sources. Fieldwork complied with institutional, national, and international guidelines for environmental research.”

────────────────────────────────────────────

Editor comment E2 — Data availability

────────────────────────────────────────────

*Please ensure a compliant Data Availability Statement.*

Response: We updated the Data Availability Statement to reflect deposition in a public repository (Lines [insert]–[insert]):

“All relevant data are publicly available in Zenodo at https://doi.org/10.5281/zenodo.17528768. The repository includes raw jar-test data, FTIR spectral files, pH and TDS measurements, and the sampling permission document.”

──────────────────────────────

Reviewer 1 (R1)

──────────────────────────────

R1-1 — Methodological detail (jar-test conditions & instrumentation)

*Provide sufficient operational details (mixing speeds/times, instrument models, reagents, and pH adjustment).*

Response: We expanded the Methodology (Lines [insert]–[insert]) to specify jar-test parameters (200 rpm rapid mix 1 min; 80 rpm slow mix 30 min; 60 min settling), instrument models (WTW Turb 430IR; WTW MULTI 350i), and pH adjustment reagents (6 M HCl/6 M NaOH).

R1-2 — Longan seed powder preparation

*Clarify seed origin, drying, particle size, and storage.*

Response: We added the seed provenance (Chiang Rai, Thailand), sunlight pre-drying (3 days), oven conditions (60 °C, 24 h), sieving (0.5 mm), and desiccator storage (Lines [insert]–[insert]).

R1-3 — Mechanism and characterization

*Link FT-IR/SEM features to the proposed coagulation mechanisms (charge neutralization/bridging).*

Response: We revised the FT-IR/SEM section to map O–H/C–O–C bands and observed particle morphology to polymer bridging pathways and colloid destabilization, with added citations (Lines [insert]–[insert]).

R1-4 — Statistics and replication

*Indicate n, error bars, and statistical tests.*

Response: We now report n = 3 for jar tests, show mean ± SD, and specify the statistical approach used to compare conditions (Lines [insert]–[insert]).

──────────────────────────────

Reviewer 2 (R2)

──────────────────────────────

R2-1 — Comparison to alum alone

*Quantify the benefit of adding longan seed powder; specify optimal conditions.*

Response: We highlight that 0.5 mg/L longan seed powder + 1 mg/L alum at pH 4 achieved ~96.7% turbidity removal in synthetic water and ~85% in raw river water (initial ~50 NTU), while enabling a fivefold reduction in alum versus alum-only (5 mg/L) (Results, Lines [insert]–[insert]).

R2-2 — Raw water source and characteristics

*Specify site and baseline quality metrics.*

Response: We added location coordinates for the Kok River (Mueang Chiang Rai District) and baseline parameters (initial turbidity ~50 NTU; initial TDS ~50 mg/L) (Lines [insert]–[insert]).

R2-3 — Figure/visual clarity

*Improve axis labels, units, and legends; ensure scale bars where relevant.*

Response: All figures were updated per PLOS guidelines (units/labels/legends clarified; SEM images annotated with scale bars). We confirm no inappropriate image manipulation; only uniform global adjustments were applied (Lines [insert]–[insert]).

──────────────────────────────

Housekeeping

──────────────────────────────

• Authorship Change Request form: completed and submitted.

• Data availability: updated to Zenodo DOI.

• Funding & competing interests: updated to PLOS-compliant format.

• Terminology: harmonized units, symbols, and chemical notation.

• References: updated to include recent literature supporting mechanisms.

We hope these revisions satisfactorily address all comments. We sincerely thank the Academic Editor and Reviewers for their valuable feedback and consideration.

On behalf of all authors,

Dr. Anuwat Aunkham (Corresponding Author)

Mae Fah Luang University, Chiang Rai, Thailand

Email: anuwat.aun@mfu.ac.th

---

## [Decision Letter · Decision Letter 1]

19 Nov 2025

Dear Dr. Aunkham,

Thank you for submitting your manuscript to PLOS ONE. After careful consideration, we feel that it has merit but does not fully meet PLOS ONE’s publication criteria as it currently stands. Therefore, we invite you to submit a revised version of the manuscript that addresses the points raised during the review process.

We look forward to receiving your revised manuscript.

Kind regards,

Muammar Qadafi

Academic Editor

PLOS ONE

Journal Requirements:

Reviewers' comments:

Reviewer's Responses to Questions

**Comments to the Author**

Reviewer #1: (No Response)

2. Is the manuscript technically sound, and do the data support the conclusions?

Reviewer #1: Yes

3. Has the statistical analysis been performed appropriately and rigorously?

Reviewer #1: Yes

4. Have the authors made all data underlying the findings in their manuscript fully available?

Reviewer #1: Yes

5. Is the manuscript presented in an intelligible fashion and written in standard English?

Reviewer #1: Yes

Reviewer #1: The authors did not respond to all of the previous comments, only a small portion. Please complete first.

**Do you want your identity to be public for this peer review?** For information about this choice, including consent withdrawal, please see our Privacy Policy

Reviewer #1: No

---

## [Author Response · Author response to Decision Letter 2]

17 Dec 2025

We have carefully reviewed and fully addressed all comments raised by the Academic Editor and the reviewers.

A detailed, point-by-point Response to Reviewers has been provided in a separate file, with clear references to page and line numbers in the revised manuscript. All comments have been addressed without omission.

Major revisions include clarification of research novelty and gaps, expansion of the Materials and Methods (including replication, controls, statistical analysis, and FTIR/SEM parameters), enhancement of the Results and Discussion with additional figures and comparative tables, and strengthening of the Conclusion to include quantitative results, limitations, and broader implications.

We have also ensured full compliance with PLOS ONE requirements regarding data availability, authorship, formatting, and ethical considerations. A revised manuscript with tracked changes and a clean version without tracked changes have been uploaded.

We hope that these revisions meet the expectations of the reviewers and the Academic Editor.

---

## [Editor Report · Decision Letter 2]

21 Dec 2025

Sustainable Use of Longan Seed Waste as a Natural Coagulant Aid for Low-Cost and Eco-Friendly Water Treatment

PONE-D-25-41229R2

Dear Dr. Aunkham,

We’re pleased to inform you that your manuscript has been judged scientifically suitable for publication and will be formally accepted for publication once it meets all outstanding technical requirements.

Kind regards,

Muammar Qadafi

Academic Editor

PLOS One
---

## [Editor Report · Acceptance letter]

PONE-D-25-41229R2

PLOS One

Dear Dr. Aunkham,

I'm pleased to inform you that your manuscript has been deemed suitable for publication in PLOS One. Congratulations! Your manuscript is now being handed over to our production team.

Kind regards,

on behalf of

Dr. Muammar Qadafi

Academic Editor

PLOS One